# Integrated Assessment of Fungi Contamination and Mycotoxins Levels Across the Rice Processing Chain

**DOI:** 10.3390/toxins17090468

**Published:** 2025-09-18

**Authors:** Carolina Sousa Monteiro, Eugénia Pinto, Rosalía López-Ruiz, Jesús Marín-Sáez, Antonia Garrido Frenich, Miguel A. Faria, Sara C. Cunha

**Affiliations:** 1LAQV-REQUIMTE, Laboratory of Bromatology and Hydrology, Faculty of Pharmacy, University of Porto, 4050-313 Porto, Portugal; carolinasousamonteiro@gmail.com (C.S.M.); mfaria@ff.up.pt (M.A.F.); 2Laboratory of Microbiology, Biological Sciences Department, Faculty of Pharmacy of University of Porto, 4050-313 Porto, Portugal; epinto@ff.up.pt; 3Interdisciplinary Centre of Marine and Environmental Research (CIIMAR/CIMAR), University of Porto, 4450-208 Matosinhos, Portugal; 4Department of Chemistry and Physics, Research Centre for Mediterranean Intensive Agrosystems and Agrifood Biotechnology (CIAIMBITAL), University of Almeria, Agrifood Campus of International Excellence, ceiA3, E-04120 Almeria, Spain; rosalia@ual.es (R.L.-R.); ms485@ual.es (J.M.-S.); agarrido@ual.es (A.G.F.)

**Keywords:** rice processing, fungi, mycotoxins, *Penicillium*, *Alternaria*, LC-MS/MS, cluster analysis, food contamination, food safety

## Abstract

This study investigated the occurrence of fungi and mycotoxins throughout the rice processing chain, from paddy rice to final white rice, in two rice varieties (variety I and variety II). A total of 75 fungal isolates were identified, belonging to the genera *Penicillium*, *Alternaria*, *Aspergillus*, *Fusarium*, and *Talaromyces*. Variety I exhibited a higher prevalence of *Penicillium* and *Alternaria*, whereas Variety II was dominated mainly by *Alternaria*, accounting for 63% of all isolates. Multi-mycotoxin screening of 22 mycotoxins revealed contamination by tenuazonic acid (TeA), zearalenone (ZEN), and 15-acetyl-deoxynivalenol (15-AcDON), with TeA concentrations exceeding 4000 µg/kg in whitened rice of variety II. Cluster analysis showed paddy and brown rice grouping together due to higher fungal loads and toxin levels, whereas whitened and final white rice clustered separately, reflecting reduced fungal counts but persistence of TeA, 15-AcDON, ZEN, and citrinin (CIT). The co-clustering of *Alternaria* with TeA and ZEN indicates strong field-related contamination. Although processing significantly decreased fungal loads, residual toxins persisted, emphasizing that rice polishing does not fully mitigate mycotoxin risks. These findings underscore the need for comprehensive surveillance and integrated management practices across the rice supply chain to minimize potential health hazards associated with fungal contaminants and their toxic metabolites.

## 1. Introduction

Rice (*Oryza sativa* L.) is one of the most important cereal crops globally, serving as the primary dietary staple for over half of the world’s population [1]. Approximately 513 million metric tons of rice were produced worldwide in 2024, with China being the leading producer [2]. Given its global importance, ensuring safety and quality across all stages of cultivation and post-harvest handling is essential for public health and food security.

Rice is a caryopsis, with the grain enclosed in a husk and composed of three main parts: the bran (pericarp and aleurone layer), the starchy endosperm (about 90% of the grain), and the germ containing the embryo and nutrients. After harvesting, the grain is referred to as paddy rice, still covered by the husk. Once de-husked, it becomes brown rice, and after further polishing to remove the bran and germ, it becomes white rice, the most widely consumed form globally [3,4].

Rice is susceptible to environmental conditions during all stages of growth. Optimal production is typically achieved in regions where daily maximum temperatures are around 28 °C and nighttime temperatures are around 22 °C. However, even a 1 °C rise in day or night temperature can cut rice yields by 7–8% [5]. As such, climate change poses a significant threat to rice productivity by increasing the frequency of abiotic stresses (such as heat and drought) and biotic stresses, including fungal infections.

The susceptibility to contamination by various filamentous fungi at different stages of its lifecycle is high, from field cultivation and harvesting to storage and industrial processing, especially under warm, humid conditions that facilitate fungal growth and toxin production. The most commonly associated genera include *Aspergillus*, *Penicillium*, *Fusarium*, and *Alternaria* [6,7]. These fungi impact grain quality and may also produce mycotoxins, toxic secondary metabolites that can persist through food processing and enter the human food chain [8].

Several mycotoxins of concern have been detected in rice and rice-derived products, including aflatoxins, ochratoxin A (OTA), citrinin (CIT), zearalenone (ZEN), and tenuazonic acid (TeA) [4,9,10,11,12,13]. These compounds are associated with a wide range of adverse health effects, such as carcinogenicity, hepatotoxicity, nephrotoxicity, immunosuppression, and endocrine disruption [12,14,15,16]. Their presence in food products is subject to strict regulation by the European Commission, which has set maximum permissible levels for several mycotoxins to protect public health [17].

Despite the importance of this issue, there is limited knowledge on how fungal contamination and mycotoxin levels evolve throughout the rice processing chain. While most studies focus on raw or final consumer rice products, fewer address the contamination dynamics from paddy rice to final white rice, or study how mycotoxins are affected by industrial milling and polishing. Previous research has shown that certain processing steps may reduce, redistribute, or even concentrate mycotoxins in specific fractions [18]. Some of these steps are critical, as improper drying, inadequate storage, and unsuitable processing practices significantly contribute to yield and nutritional quality losses [19]. Post-harvest losses can account for 15–16% of total rice production in developing countries, and in extreme cases, may reach up to 40–50% [20]. Additionally, overall crop loss due to pests and pathogens, including mycotoxigenic fungi, has been estimated at approximately 37% globally for rice [21].

In this context, combining fungal isolation and identification, using both traditional culturing and molecular barcoding, with multi-mycotoxin quantification by Ultra-high performance liquid chromatography–tandem mass spectrometry (UHPLC-MS/MS) provides a comprehensive approach to understanding the evolution of contamination throughout the rice supply chain. It is especially relevant in Europe, where regulatory frameworks demand high analytical precision and traceability for food safety compliance.

This study aims to investigate the presence and diversity of mycotoxigenic fungi, such as *Aspergillus*, *Penicillium*, *Fusarium*, and *Alternaria*, and quantify multiple mycotoxins across key stages of the rice processing chain, from paddy rice to the final white rice product. By integrating microbiological, molecular, and chemical analyses, this research provides valuable data to support risk assessment, regulatory compliance, public health protection, and reinforces food security frameworks, particularly in vulnerable regions and Europe, where rice is both consumed and imported in large volumes.

## 2. Results and Discussion

### 2.1. Method Validation

Matrix effects were assessed by comparing the slopes of calibration curves prepared in pure solvent with those of matrix-matched standards for each matrix. For some mycotoxins, pronounced matrix effects were observed. To address the matrix effects and ensure accurate quantification, all mycotoxins were analyzed using matrix-matched calibration standards in combination with a stable isotope-labeled internal standard (OTA-d5). This combined approach effectively compensated for the matrix-induced variability, enhancing quantification accuracy and robustness. Matrix-matched calibration curves were established by spiking blank extracts at six or more concentration levels. All analytes exhibited satisfactory linearity, with a coefficient of determination (R^2^) exceeding 0.91 (Appendix A).

Accuracy and precision were verified through recovery experiments performed by spiking blank samples at three concentration levels, each tested in five replicates. Average recoveries across all tested levels ranged from 70% to 120%, aligning well with the performance requirements set out in EU guidelines (EC, 2006b) [22]. Repeatability and intermediate precision, expressed as intra-day and inter-day relative standard deviation (RSD, %), did not exceed 20%, demonstrating the method’s reliability for routine multi-mycotoxin analysis (Appendix A).

Carryover was assessed by injecting a blank solvent immediately after the highest calibration standard. No detectable peaks corresponding to the monitored mycotoxin transitions were observed, confirming the absence of residual analyte carryover.

The analytical method exhibited excellent sensitivity for most mycotoxins. Limits of detection (LOD) and quantification (LOQ) were defined as the lowest concentrations yielding signal-to-noise (S/N) ratios of 3 and 10, respectively (Appendix A). LODs ranged from 0.1 to 2.0, while the LOQs ranged from 0.3 to 4.5 µg/kg, demonstrating the method’s capability to detect and quantify mycotoxins at concentrations well below the maximum residue limits established by the European Commission for processed cereal-based foods [17,22,23,24,25].

For the quantification of paddy rice, brown rice, and rice bran, brown rice matrix-matched calibration standards were used. For whitened rice, polished rice, broken rice, and final white rice, white rice matrix-matched calibration standards were used.

### 2.2. Occurrence and Distribution of Mycotoxins Along the Rice Processing Chain

Fungal contamination and mycotoxin accumulation in rice are complex, multifactorial processes influenced by environmental conditions, storage practices, and industrial processing. While extensive research has been conducted on mycotoxins in cereals such as wheat and maize, studies specifically examining how fungal presence and mycotoxin levels evolve throughout the entire rice processing chain remain limited [6,7,8].

For two rice varieties, samples of paddy rice, brown rice, bran rice, whitened rice, polished rice, and the final white rice were screened for 22 mycotoxins. Quantification results for all types of samples are depicted in Table 1. The analysis revealed a diverse pattern of contamination highlighting the impact of processing steps on the prevalence and concentration of mycotoxins. All the samples under study tested positive for different mycotoxins, and co-occurrence was observed with a minimum of five and a maximum of eleven mycotoxins present. Samples from rice variety II are more contaminated than those from variety I, suggesting different susceptibilities between them.

Paddy rice samples contained a relatively high diversity of mycotoxins, with seven distinct mycotoxins detected across both rice varieties. These included notable levels of 15-acetyl-deoxynivalenol (15-AcDON) (up to 113.28 µg/kg), TeA (up to 2121.37 µg/kg), beauvericin (BEA) (up to 22.99 µg/kg), CIT (22.57 µg/kg), enniatins B1(ENNB1) (18.21 µg/kg), ZEN (up to 62.98 µg/kg), and sterigmatocystin (STG) only in variety I. TeA dominated the profile, particularly in variety II, highlighting field contamination as the likely primary source for this *Alternaria* toxin. These results highlight the susceptibility of unprocessed rice to field-associated fungi such as *Fusarium* and *Alternaria*, which are known producers of trichothecenes, ZEN, and TeA, respectively [4]. The initial contamination levels establish a critical baseline, indicating that significant toxin loads can already be present before any processing.

Brown rice, which results from de-husking but retains the bran layer, is often marketed as a healthier alternative due to its higher content of dietary fiber, vitamins, and minerals [3]. Despite its nutritional benefits, brown rice showed significant contamination, with up to seven mycotoxins detected across both varieties, including high levels of TeA (2395.11 µg/kg) and CIT (91.61 µg/kg) in variety II. Levels of 15-AcDON remained elevated (56.21 to 83.75 µg/kg), while BEA and ZEN were consistently detected across both varieties. Notably, brown rice contained more mycotoxins overall than paddy rice in variety I, highlighting how the removal of the husk does little to mitigate contamination within the outer grain layers. It suggests that the bran layer, which remains intact in brown rice, serves as a critical reservoir for mycotoxins [4,26].

Rice bran, a by-product of the polishing process, was consistently the most contaminated fraction, with the highest number (variety I: 6; variety II: 11) and concentration of mycotoxins detected, especially in variety II. This included exceptionally high levels of 15-AcDON (up to 965.23 µg/kg), TeA (up to 5419.88 µg/kg), BEA (up to 159.19 µg/kg), CIT (152.23 µg/kg), ZEN (up to 220.45 µg/kg), as well as 3-AcDON, DON, ENNB, ENNB1, STG, and T-2. These concentrations reflect the known accumulation of fungal toxins in the aleurone and pericarp layers [4,26]. From a food safety perspective, this highlights the importance of rigorous monitoring for rice bran products, which are increasingly incorporated into functional foods and animal feed.

Whitened rice, an intermediate product before complete polishing, still showed moderate levels of contamination, with up to nine mycotoxins detected in variety II and five in variety I. Key findings included TeA (4264.81 µg/kg), 15-AcDON (35.86 µg/kg), BEA (19.16 µg/kg), and CIT (50.95 µg/kg). The persistence of these toxins demonstrates that although mechanical abrasion removes some contaminated layers, considerable residues remain. Notably, OTA was first detected at this stage in variety II (34.44 µg/kg). While generally reduced compared to bran, the high TeA levels in whitened rice are of particular concern, as they indicate this emerging toxin is not substantially reduced by initial milling.

Further processing to polished rice generally resulted in lower mycotoxin concentrations, but significant residues were still present, with eight different mycotoxins detected in variety II and six in variety I. TeA levels were notably lower (2219.83 µg/kg) compared to bran and whitened rice but remained substantial. Additionally, 15-AcDON (124.25 µg/kg), BEA (37.57 µg/kg), CIT (50.10 µg/kg), ENNB (4.19 µg/kg), ENNB1 (45.62 µg/kg), ZEN (17.35 µg/kg), and OTA (34.68 µg/kg) persisted. This indicates that while polishing significantly decreases contamination, likely by removing additional layers of the aleurone and sub-aleurone tissues, certain mycotoxins, especially TeA, 15-AcDON, and OTA, can still be retained within the endosperm or result from cross-contamination.

The final consumer product, white rice, exhibited the lowest contamination levels in both varieties, with five to seven mycotoxins still detectable. For rice variety I, TeA decreased to 51.30 µg/kg, 15-AcDON remained around 19.04 µg/kg, and ZEN dropped to 9.10 µg/kg. In contrast, variety II showed a more concerning profile, with TeA at 407.76 µg/kg, 15-AcDON at 95.74 µg/kg, and residual OTA still present. However, the final white rice conforms to the EU maximum limits [17,22,23,24,25].

Few studies have assessed the mycotoxin content throughout the rice processing chain. A survey of stored paddy rice revealed a high prevalence of ZEN (30%), FBs (10%), BEA/AFB2 (6%), and, similar to our study, a high incidence of co-occurrence of mycotoxins [27]. Usually, the reports focus on the final product that is ready for consumption. Contrary to our study, the most prevalent mycotoxins at this stage are AFs (0.03–371.9 µg/kg), BEA (0.2–57.4 µg/kg), ENNs (0.06–12.68 µg/kg), and FBs (0.4–176.58 µg/kg). Moreover, the highest mean concentrations were observed for nivalenol (170.13 μg/kg), DON (56.13 μg/kg), FB1 (50.29 μg/kg), zearalenone (38.55 μg/kg), and OTA (18.64 μg/kg) [8,28,29].

Figure 1 presents the average concentrations of mycotoxins detected in both rice varieties, enabling a direct comparison between two commonly consumed rice types, browns and white. Brown rice consistently had higher levels of all mycotoxins, as it still contains the bran layer where many mycotoxins concentrate. Processing into white rice achieved substantial reductions, most notably for CIT (91.40%), TeA (81.20%), ZEN (80.0%), and BEA (75.80%), while 15-AcDON decreased by only 18.0%, indicating that it remains relatively persistent even after polishing. These findings emphasize the trade-off between the nutritional benefits of brown rice and increased exposure to mycotoxins. While nutritionally richer, it may pose greater risks of mycotoxin exposure, underscoring the need for specific monitoring and consumer awareness, especially under evolving safety frameworks. Processing reduced overall contamination, particularly for bran-associated mycotoxins such as ZEN and trichothecenes. The persistence of TeA, OTA, and *Fusarium* toxins in final white rice highlights the limitations of mechanical processing alone in eliminating contamination risks.

Overall, the mycotoxins decrease during the processing chain, with the final white rice always having a lower amount than the initial paddy rice, although some steps can enhance them. The evolution of the detected mycotoxins through the rice processing steps in both varieties is described in Figure 2.

Among the trichothecenes analyzed, 15-AcDON was consistently the most prevalent. For both rice varieties, the highest concentrations were observed in rice bran, reaching 591.58 µg/kg and 965.23 µg/kg for variety I and II, respectively. This aligns with the literature, which indicates that milling by-products, such as bran, frequently accumulate higher levels of *Fusarium* mycotoxins due to the removal of outer grain layers, where these toxins typically concentrate [18]. Milling reduced 15-AcDON levels by 96–98%, though complete elimination was not achieved. These findings highlight the role of bran separation in reducing dietary exposure to these compounds. Yet, they also demonstrate that white rice still contributes to intake, which is relevant under the EU maximum limit for DON (including acetylated forms when converted) of 1000 µg/kg for unprocessed cereals and 75 µg/kg for cereals intended for consumption (except rice) [23]. 3-AcDON was detected in bran in variety II, and entirely removed by the milling process. DON was only detected in the rice bran of variety II and was absent in all other stages and variety I. This indicates specific contamination isolated to bran, consistent with DON’s known localization in outer grain tissues. Thus, milling here effectively eliminated DON from white rice [11].

ENNB and ENNB1 appeared mainly after milling and whitening, suggesting these emerging *Fusarium* toxins can survive processing. The same was observed with BEA, with notably high levels in rice variety II, particularly in rice bran (159.19 µg/kg) and still considerable concentrations in polished rice (37.57 µg/kg), which were reduced by 80% after separation of the broken rice.

CIT presented a striking difference between varieties. In variety I, CIT was almost absent, detected only in brown rice, possibly reflecting minor surface contamination. While in variety II, it began at 22.57 µg/kg in paddy rice and quadrupled in brown rice. It peaked in bran, indicating concentration in the outer grain layers. Subsequent steps lowered the CIT to approximately 50 µg/kg in whitened, polished, and broken rice, representing a 67% decrease from bran and an 84% decrease after separating the broken fraction, reaching 7.99 µg/kg in final white rice. This confirms that CIT primarily concentrates in bran layers and is reduced by standard milling, although it is not eliminated. Given that citrinin is regulated only in food supplements (with an EU limit of 100 µg/kg), its presence in rice suggests that indirect exposure routes warrant monitoring.

OTA appeared uniquely in variety II, specifically in whitened, polished, and broken rice, at 34–37 µg/kg, and was completely absent in paddy, brown, bran, and final white rice. This unusual pattern suggests possible post-harvest or storage contamination, with OTA-producing *Aspergillus* growing on semi-processed rice. It is concerning because OTA exceeded the EU limit of 3 µg/kg, highlighting the importance of maintaining optimal storage conditions.

TeA was consistently the most abundant mycotoxin in both rice varieties, though its behavior across processing stages varied significantly. In rice variety I: TeA levels start at 292.1 µg/kg in paddy rice, and they dropped by 85% to 42.94 µg/kg in brown rice (after husk removal). However, concentrations increased approximately 21-fold in rice bran (917 µg/kg), illustrating how milling concentrates TeA in the outer layers of the grain. The milling resulted in a 78% increase in whitened rice and a subsequent 64% decrease in polished rice. Interestingly, broken rice showed an unexpected increase to 323.59 µg/kg, suggesting possible redistribution or selective contamination of broken fractions. The final white rice, after selection from broken rice, contained only 51.3 µg/kg, approximately 36% lower than polished rice. The load decrease between the initial and final product is 83%. In variety II, the contamination profile was more severe. TeA levels are already high in paddy (2121.37 µg/kg). TeA increased slightly, by 13%, in brown rice, and then more than doubled concentrated in rice bran, reaching 5419.88 µg/kg. Even after polishing, TeA levels remained high, increasing 78% from the brown rice to whitened. Subsequent polished steps, decreased 48% and 82% to final white rice. This highlights that milling concentrates TeA in the bran fraction, but redistribution into broken and lower-grade polished fractions can also occur. Although final white rice showed significant reductions, it still retained concerning levels, particularly in variety II. Such levels significantly exceed those commonly reported for cereals [30,31], though comparable levels have been observed on other food products [32,33].

ZEN followed a similar pattern in both varieties with the highest concentration in bran, and a decrease through polishing. Milling reduced ZEN by 14% and 85% (variety I and II) compared brown rice, but residual levels were still measurable in all final white rice. Another study reported a reduction through milling [11]. Although the EU has not established maximum limits for ZEN in rice, the levels in some samples exceeded the thresholds set for other cereals, 100 µg/kg (for unprocessed grains) and 75 µg/kg (for grains intended for consumption).

Interestingly, AFB2 was detected only in broken rice of variety II, highlighting localized contamination, possibly due to fungal colonization following kernel fracture. This also underscores why broken rice often has higher safety risks.

STG was generally detected at low levels, primarily in early and by-product stages, with a tendency to concentrate in bran. Additionally, T-2 was detected after milling, in bran and whitened rice of variety II, and was absent from the final white rice, indicating a sporadic occurrence.

The data demonstrate that milling and polishing significantly reduce mycotoxin levels in white rice, confirming their effectiveness in mitigating consumer exposure. However, the accumulation of mycotoxins in by-products, particularly rice bran, raises serious concerns, especially as these materials are used in animal feed or processed foods [34].

Across both rice varieties, the results confirmed that rice bran systematically contained the highest concentrations and the most diverse array of mycotoxins, followed by broken and whitened rice. Final white rice typically showed an 80–90% reduction in mycotoxin levels compared to bran. However, residual contamination was still common, especially for 15-AcDON, BEA, CIT, TeA, and ZEN, underscoring that milling alone does not guarantee complete safety. Importantly, OTA was detected in semi-processed rice at levels exceeding EU regulatory limits, raising concerns about post-harvest contamination. The concentration of emerging mycotoxins, such as TeA, BEA, and ENNs, which are not yet regulated but biologically active, emphasizes the need for holistic monitoring and stricter quality control throughout the entire processing chain. These findings underscore the importance of integrated strategies combining field-level fungal control, optimized storage conditions, and targeted monitoring at each processing stage to ensure food safety.

### 2.3. Mycoflora Isolation and Identification

Despite the application of good agricultural practices and postharvest processing measures, complete decontamination of food products remains unattainable. Fungi are remarkably resilient contaminants due to their adaptability to diverse environmental conditions and their ability to produce mycotoxins in response to stress [4]. In this work, fungal isolation was performed in key processing steps, paddy, brown, and whitened, leading to the final white rice. The identified species, as determined using the morphological and molecular techniques described, are listed in Table 2 and complemented by the phylogenetic trees (Appendix A). The fungal diversity observed across the rice processing chain in both varieties highlights the persistence and variability of fungal contamination, despite successive processing steps aimed at reducing microbial loads. In rice variety I, a total of 48 fungal isolates were obtained, with *Penicillium* spp. being the most prevalent genus (41.7%), followed by *Alternaria* spp. (27.1%) and *Aspergillus* spp. (20.8%). The presence of *Fusarium* and *Talaromyces* was residual. In contrast, rice variety II exhibited a distinct fungal profile, comprising a total of 27 fungal isolates, with *Alternaria* spp. being the dominant species. (63.0%). *Aspergillus* spp. is ranked in the second position, whereas *Penicillium* spp. were rarely detected (7.4%), contrasting sharply with their prominence in variety I. The presence of *Fusarium* and *Talaromyces* was again residual.

Generally, *Alternaria* spp. was the most prevalent genus overall, followed by *Penicillium* spp., and *Aspergillus* spp. *Fusarium* spp. and *Talaromyces* spp. were minorly isolated fungi, suggesting tolerance to low water activity and processing conditions. The species composition identified in this study broadly agrees with earlier surveys of rice mycobiota, which often cite *Alternaria*, *Aspergillus*, *Penicillium*, and *Fusarium* as the dominant genera associated with rice grains during cultivation, storage, and processing [4,11,35,36,37]. However, the contrasting dominance of *Penicillium* in variety I versus *Alternaria* in variety II highlights the influence of varietal susceptibility, field microecology, and possibly differing post-harvest handling practices [38]. Such variation emphasizes the need for targeted monitoring and tailored interventions based on rice type and processing flow.

In both rice varieties, paddy rice samples showed high fungal diversity, with 25.0% (variety I) and 37.0% (variety II) of total isolates. The dominant genera were *Alternaria,* a common contaminant of rice in the field and during early storage stages, due to its ability to grow at low temperatures and moisture levels typical of harvested grains [4]. Notably, *Fusarium* species were isolated from paddy in both varieties (*F. fujikuroi* in variety I, *F. tanahbumbuense* in variety II), reflecting primary infections likely acquired pre-harvest [4,39]. *Penicillium* and *Aspergillus* species were also isolated, although present in the field, they are frequently associated with post-harvest practices and storage conditions [35,40].

Brown rice also exhibited substantial fungal colonization, 29.2% (variety I) and 37.0% (variety II) of isolates. This stage, which retains the bran layer, continued to support high loads of *Alternaria* (particularly *A. alternata* and *A. destruens*), especially in variety II. *Penicillium* was more frequent in variety I. The persistence of *Fusarium* in variety II underscores that ordinary de-husking alone is insufficient to eliminate field fungi.

After initial milling, whitened rice samples showed the highest fungal load in variety I (33.3%), primarily *Penicillium* and *Alternaria*, followed by *Aspergillus* and *Talaromyces*. In variety II, the fungal load decreased to 22.2%, with *Alternaria* still prominent. This suggests that partial removal of outer layers reduces, but does not eliminate, the presence of surface-associated fungi.

In fully polished rice, fungal isolates decreased markedly to 12.5% (variety I) and only 3.7% (variety II). While *Alternaria*, *Aspergillus*, and occasional *Penicillium* were still detected, their reduced presence suggests that polishing significantly lowers fungal contamination. Notably, the persistence of toxigenic genera into the final white rice, although at reduced frequencies, raises concerns given their well-established capacity to produce harmful mycotoxins, and highlights potential risks for consumer exposure, particularly when storage conditions allow residual fungal populations to proliferate [36]. Remarkably, *P. charlesii*, *A. resedae,* and *A. alternata* predominated throughout the processing chain until the whitened rice stage, suggesting their resilience or even possible proliferation during intermediate polishing steps. *Alternaria* species, including *A. alternata* and *A. resedae*, reflect typical field-origin contamination that can withstand initial de-husking. The detection of *Fusarium* species was mainly in the initial steps, consistent with its role as a typical field fungus less adapted to the conditions prevailing in processed fractions [11,39]. The detection of *F. fujikuroi* in both paddy and final white rice samples may indicate sporadic contamination events or the survival of the fungus in protected grain microenvironments. The presence of *Talaromyces* species was limited, but it mainly appeared in later processing stages.

Hierarchical clustered heatmaps for both rice varieties (Figure 3) provide a comprehensive visualization of contamination profiles across the rice processing stages, integrating the occurrence of fungal genera with mycotoxin concentrations. The plots were carried out using the pheatmap package in R (https://cran.r-project.org/web/packages/pheatmap/index.html (10 July 2025)). Each cell indicates the degree to which a given fungus or mycotoxin deviates from the overall mean across samples, with red indicating higher values and blue indicating lower values. For Variety I, clustering revealed that whitened rice and final white rice grouped together, indicating similar contamination profiles characterized by relatively lower levels of most fungi and mycotoxins, except for moderate persistence of *Alternaria* and residual levels of 15-AcDON and ENNB. In contrast, paddy rice and brown rice clustered separately, driven by higher abundances of *Penicillium* and *Alternaria*, as well as elevated levels of TeA, ZEN, and 15-AcDON, reflecting contamination patterns that largely originated from the field and early processing stages. The clustering of variables distinguished a group primarily composed of field and storage-associated fungi (e.g., *Penicillium*, *Alternaria*), which produce mycotoxins such as TeA and ZEN, suggesting potential co-occurrence. In contrast, Talaromyces appeared more aligned with the final processing stages. For Variety II, a similar dichotomy was observed, with a distinct pattern before and after milling that emphasized the impact of the processing. Paddy rice and brown rice form one cluster, marked by high levels of *Alternaria*, TeA, CIT, and ZEN, consistent with initial field contamination. Conversely, whitened rice and final white rice were grouped together, showing reduced fungal loads but notable persistence of certain mycotoxins such as OTA and ENNB1, and unique clustering with *Aspergillus*. These patterns indicate that while processing effectively reduces many fungal load, certain toxigenic fungi and associated mycotoxins may persist or even become relatively more prominent in later stages.

These clustered profiles highlight the complexity of contamination dynamics along the rice processing chain and serve as a tool to identify critical control points and potential co-regulatory contamination patterns that may not be apparent from isolated prevalence or concentration data. This work emphasizes that while mechanical processing steps substantially reduce the fungal load, they do not uniformly eliminate associated mycotoxins, as these toxins have stable structures and may persist from the farm to the plate, ultimately reaching the consumer.

## 3. Conclusions

The data indicate that rice bran consistently exhibited the highest mycotoxin load, followed by whitened and polished rice. Rice bran, though not directly consumed, serves as a source of nutrients and bioactive compounds; therefore, its high contamination levels are a significant concern. On the other hand, the most consumed white rice generally contained the lowest levels of mycotoxins; however, the number and variety of mycotoxins detected across processing stages highlight the widespread and multi-toxin nature of contamination.

Of particular concern is the consistently high presence of TeA, an *Alternaria* toxin not yet regulated in the EU but increasingly recognized for its cytotoxic and genotoxic potential. Similarly, OTA levels detected in the polished and whitened rice fractions of variety II exceeded the EU maximum limit of 3 µg/kg for cereals, posing potential regulatory non-compliance and health risks.

This study highlights the persistent and complex nature of fungal and mycotoxin contamination along the rice processing chain. Our findings revealed that paddy and brown rice exhibited the highest levels of fungal contamination, dominated by *Penicillium* and *Alternaria*, along with substantial concentrations of TeA, ZEN, and 15-AcDON. While milling and polishing significantly reduce fungal loads and mycotoxin concentrations, they do not eliminate key contaminants, such as TeA, 15-AcDON, OTA, ZEN, and CIT, which persist in whitened and final white rice. The distinct clustering patterns observed in heatmaps reinforce that mycotoxin profiles are closely linked to fungal occurrence, with strong relationships between *Alternaria* and field-origin mycotoxins, such as TeA and ZEN. Importantly, even polished rice samples contained residual fungi and mycotoxins, emphasizing that processing alone does not guarantee safety. Given rice’s global dietary significance, these results underscore the urgent need for integrated control strategies that combine field management, optimized storage, and targeted screening of both fungi and toxins throughout the entire processing chain. Future work should expand sampling across diverse production systems and incorporate quantitative risk assessments to better protect consumers. Collectively, our study provides critical insights into the dynamic behavior of fungal contamination and mycotoxin accumulation in rice, supporting efforts to strengthen food safety monitoring and regulatory frameworks.

## 4. Materials and Methods

### 4.1. Reagents and Standard Solutions

Certified mycotoxin standards with purity ≥95% for 3-acetyl-deoxynivalenol (3-AcDON), 15-acetyl-deoxynivalenol (15-AcDON), HT-2 toxin (HT-2), T-2 toxin (T-2), aflatoxin B2 (AFB2), aflatoxin G1 (AFG1), aflatoxin G2 (AFG2), zearalenone (ZEN), tenuazonic acid (TeA), beauvericin (BEA), and enniatins A, A1, B, and B1 (ENNA, ENNA1, ENNB, ENNB1) were obtained from Sigma-Aldrich (West Chester, PA, USA). Standards of deoxynivalenol (DON), citrinin (CIT), and cyclopiazonic acid (CPZ), with purity ≥98%, were purchased from Toronto Research Chemicals (TRC, Toronto, ON, Canada). Aflatoxin B1 (AFB1), fumonisin B1 (FB1), and fumonisin B2 (FB2) standards (purity ≥98%) were supplied by LGC (Teddington, Middlesex, UK). Ochratoxin A (OTA, ≥98% purity) was sourced from Santa Cruz Biotechnology (Heidelberg, Germany), and sterigmatocystin (STG, ≥98% purity) from BioViotica (Liestal, Switzerland). Isotopically labeled internal standards (IS) included OTA-d5 and ^13^C_18_-STG, purchased from TRC. High-performance liquid chromatography (HPLC)-grade methanol (MeOH), acetonitrile (ACN), acetic acid, and formic acid were obtained from Merck (Darmstadt, Germany), along with analytical-grade ammonium acetate. Ultrapure water was produced using a SeralPur Pro 90 CN purification system (SERAL, Ransbach-Baumbach, Germany). Acquity UPLC BEH C18 (100 × 2.1 mm, 1.7 µm) was sourced from Waters (Milford, MA, USA). Anhydrous magnesium sulfate (MgSO_4_) was purchased from Sigma and pre-treated at 500 °C for 5 h before use. Stock standard solutions of individual mycotoxins (100 mg/L) were prepared in ACN/MeOH. Internal standards OTA-d5 (500 mg/L) and ^13^C_18_-STG (25 mg/L) were prepared in DMSO and ACN, respectively. All solutions were stored at −18 °C until use.

Sabouraud Dextrose Broth (SDB) and Sabouraud Dextrose Agar (SDA) were obtained from Biomérieux (Lyon, France). Czapek Yeast Autolysate Agar (CYA) was purchased from Becton Dickinson (Franklin Lakes, NJ, USA), and yeast extract was sourced from Conda Lab (Madrid, Spain). Magnesium sulfate heptahydrate was obtained from Merck (Darmstadt, Germany), agar from Liofilchem (Téramo, Italy), and chloramphenicol from Sigma-Aldrich (St. Louis, MO, USA).

### 4.2. Sampling

An industrial rice processing company supplied two distinct varieties of rice. Representative samples were collected at each stage of the processing chain, including paddy rice, brown rice, rice bran, whitened rice, polished rice, broken rice, and the final consumer product (hereafter referred to as final white rice). Some of the samples were ground using a Vorwerk Bimby TM6 (Vorwerk, Wuppertal, Germany) for mycotoxin screening, and another part was saved whole for mycoflora isolation and identification. All samples were stored under vacuum-sealed conditions in a cold, dry, and dark environment to preserve their integrity until analysis.

### 4.3. Determination of Mycotoxins by UHPLC-MS/MS

#### 4.3.1. Extraction Protocol

Mycotoxins were extracted using a modified QuEChERS-based method, adapted from previously established protocols [41,42]. Briefly, 1 g of homogenized sample was placed into a 50 mL centrifuge tube, followed by the addition of 40 µL of OTA-d5 (1 mg/L) as an internal standard (IS1). The mixture was shaken using a mechanical shaker for 1 h. Subsequently, 5 mL of acetonitrile (ACN) containing 5% formic acid was added, and the mixture was shaken overnight. Afterward, 5 mL of ultrapure water, 2 g of anhydrous magnesium sulfate (MgSO_4_), and 0.5 g of sodium chloride (NaCl) were added. The tube was vortexed vigorously for 1 min and then centrifuged at 8497× *g* for 10 min. An aliquot of 1 mL from the upper organic phase was collected and evaporated to dryness under a gentle stream of nitrogen. The residue was reconstituted in 720 μL of mobile phase B, and 30 μL of ^13^C_18_-STG (1 mg/L) was added as a second internal standard (IS2). A 10 µL volume of the final extract was injected into the UHPLC-MS/MS system for analysis. Each sample was extracted in duplicate, and each extract was injected twice.

#### 4.3.2. Instrument and Analytical Conditions

Mycotoxin analysis was performed using an Agilent 1290 Infinity Rapid Resolution Liquid Chromatography (RRLC) system (Agilent Technologies, Santa Clara, CA, USA), equipped with a thermostated autosampler (G1330B), binary pump (G4220A), and a thermostated column compartment (G1316C). The UHPLC system was coupled to an Agilent 6460A triple quadrupole mass spectrometer (QqQ) with a Jet Stream electrospray ionization (ESI) source (G1958-65138). Chromatographic separation was achieved on an Acquity UPLC BEH C18 column (100 × 2.1 mm, 1.7 µm) maintained at 28 °C. Using a gradient elution program, the mobile phase was delivered at a flow rate of 300 µL/min. Mobile phase A consisted of water/methanol/acetic acid (94:5:1, *v*/*v*) with 5 mM ammonium acetate, and mobile phase B consisted of methanol/water/acetic acid (97:2:1, *v*/*v*) with 5 mM ammonium acetate. The gradient profile was as follows: 0–3.5 min, 92% A; 5.0–6.0 min, 35% A; 6.0–7.0 min, 25% A; 7.0–10.5 min, 10% A; 10.5–11.0 min, 25% A; and 11.0–15.0 min, 92% A.

The MS conditions were as follows: gas flow, 5 L/min; source temperature, 325 °C; sheath gas temperature, 400 °C; nebulizer pressure, 45 psi; sheath gas flow, 11 L/min; capillary voltage, positive mode, 3500 V. The cycle time for dynamic multiple reaction monitoring (dMRM) parameters was 500 ms. Nitrogen was used as a nebulizing and collision gas. The data acquisition was performed using dMRM mode, employing ESI in both positive and negative modes within the same run. For each analyte, between two and four transitions were selected for identification, and the corresponding cone voltage and collision energy were optimized for maximum intensity. The optimized MS/MS parameters for the target analytes are listed in the Appendix A. Data acquisition and processing were carried out using MassHunter software, version 10.0 (Agilent Technologies, Santa Clara, CA, USA).

#### 4.3.3. Method Validation and Quality Control

White rice and brown rice matrices were used for the method validation (Appendix A). Linearity was assessed through calibration curves constructed using at least six concentration levels ranging from 2.5 to 160 µg/kg for AFB1, AFB2, AFG1, AFG2, BEA, CIT, CPZ, ENNA, ENNA1, ENNB, ENNB1, HT-2, OTA, STG, T-2, ZEN, and TeA, and from 12.5 to 600 µg/kg for 15-AcDON, 3-AcDON, DON, FB1, FB2. Method accuracy and precision were assessed through recovery experiments. Precision was expressed as the relative standard deviation (RSD) for both intra-day and inter-day repeatability. These parameters were evaluated by analyzing spiked samples (*n* = 5) at three concentrations, following the guidelines of the European Commission (EC) (EC 2006b) [22]. Blank rice samples (1 g) were fortified after homogenization with 40 µL of OTA-d5 (1 mg/L (IS1)). The spiked samples were shaken on a mechanical shaker for 1 h, and then 5 mL of ACN with 5% formic acid was added. The mixture was equilibrated overnight and processed as described in the “Extraction Protocol” section. Limits of detection (LODs) and quantification (LOQs) were defined as the lowest analyte concentrations producing signal-to-noise (S/N) ratios of 3 and 10, respectively (Appendix A).

### 4.4. Mycoflora Isolation and Identification

#### 4.4.1. Fugal Isolation

Fungal isolation was conducted on paddy rice, brown rice, whitened rice, and the final white rice. These stages were selected to assess changes in fungal presence across critical processing steps leading to the consumer-ready product. By-products and further refined fractions not directly intended for consumption, such as rice bran, polished rice, and broken rice, were excluded from this analysis. Three fungal culture media were used for isolation: Sabouraud Dextrose Agar (SDA), Czapek Yeast Autolysate Agar (CYA), and Yeast Extract Sucrose Agar (YES), each supplemented with chloramphenicol to inhibit bacterial growth.

Depending on the rice matrix, two approaches were employed for fungal isolation [43,44,45]. Direct plating: Seven rice grains from each sample were placed at equidistant points on the surface of Petri dishes containing each culture medium. Wash solution method: Five grams of rice were macerated in 45 mL of sterile 0.1% peptone solution and shaken for 45 min. After seven post-washed rice grains of each sample were dropped as described above, plus 100 µL of either diluted or concentrated wash solution was spread on the agar surface. Concentration was achieved by centrifugation at 8497× *g* for 5 min, and the supernatant was reduced to the required final volume (paddy rice diluted 1:2; brown rice concentrated to 20 mL; whitened rice and final white rice concentrated to 5 mL). The isolation method varied by sample type to adapt to expected contamination levels (higher in paddy rice and lower in final white rice), prevent overgrowth or false negatives, and optimize detection sensitivity and specificity across different rice types and processing stages. All inoculated plates were incubated at 25 °C in the dark for five days. After incubation, microbial growth was assessed and classified based on the presence of bacteria, yeasts, or filamentous fungi. All isolated fungal strains were stored in SDB with 20% glycerol at −80 °C. Before subsequent analysis, isolates were sub-cultured on SAB agar to confirm viability and purity.

#### 4.4.2. Morphological Identification

All the fungal colonies obtained in the culture plates were isolated for further identification. The strains of *Aspergillus*, *Penicillium*, *Fusarium,* and *Alternaria* were further identified using conventional morphological techniques, based on their macroscopic and microscopic characteristics when growing in different media, as reported in the literature [43,46].

#### 4.4.3. Molecular Identification

Morphologically identified isolates belonging to the genera *Aspergillus*, *Penicillium*, *Fusarium*, and *Alternaria* were further characterized at the molecular level. Genomic DNA was extracted using the E.Z.N.A. Plant & Fungal DNA Kit (Norcross, GA, USA) according to the manufacturer’s protocol. DNA extracts were dissolved in elution buffer and stored at 4 °C until further use. Genus-level identification was performed by amplifying the ribosomal internal transcribed spacer (ITS) region using the universal fungal primers ITS1 (5′-TCC GTA GGT GAA CCT GCG G-3′) and ITS4 (5′-TCC TCC GCT TAT TGA TAT GC-3′). Two additional barcode regions were amplified for each genus to achieve species-level identification. For *Aspergillus* and *Penicillium* was used β-tubulin (Bt) gene using the primers Bt2a/Bt2b (5′-GGT AAC CAA ATC GGT GCT GCT TTC-3′/5′-ACC CTC AGT GTA GTG ACC CTT GGC-3′) and calmodulin (CaM) gene using the primers cdm5/cdm6 (5′-CCG AGT ACA AGG ARG CCT TC-3′/5′-CCG ATR GAG GTC ATR ACG TGG-3′); For *Fusarium* targeting the translation elongation factor 1-alpha (EF-1α) gene using the primers EF1/EF2 (5′-ATG GGT AAG GAR GAC AAG AC-3′/5′-GGA RGT ACC AGT SAT CAT GTT-3′) and RNA polymerase II (RPB2) gene using primers rPB2_5f2/Rpb2_7cr (5′-GGG GWG AYC AGA AGA AGG C-3′/5′-CCC ATR GCT TGY TTR CCC AT-3′); for *Alternaria* was used gene encoding the plasma membrane ATPase using the primers ATPDF1/ATPDR1 (5′-ATC GTC TCC ATG ACC GAG TTC G-3′/5′-TCC GAT GGA GTT CAT GAT AGC C-3′) and targeting the Alternaria major allergen (alt) using the primers alt a 1F/alt a 1R (5′-ATG CAG TTC ACC ACC ATC GC-3′/5′-ACG AGG GTG AYG TAG GCG TC-3′) [43,46,47,48,49,50,51,52,53]. Polymerase chain reactions (PCR) were performed using a T100™ Thermal Cycler (Bio-Rad, Irvine, CA, USA) in a total volume of 24 µL. Each reaction contained: 1× FIREPol^®^ buffer, 2.0 mM MgCl_2_, 0.2 mM of each dNTP, 0.3 µM of each primer, 1 U of FIREPol^®^ Taq polymerase (Solis BioDyne, Tartu, Estonia), and 1 µL of genomic DNA (~10 ng). The thermal cycling conditions were as follows: initial denaturation at 95 °C for 15 min; 40 cycles of denaturation at 95 °C for 20 s, annealing at target-specific temperatures (ITS: 50 °C; Bt: 56 °C; CaM: 58 °C; EF: 52 °C; RPB2: 55 °C; ATP: 59 °C; Alt a 1: 57 °C) for 30 s, and extension at 72 °C for 1 min); followed by a final extension at 72 °C for 2 min. A no-template negative control was included in each batch of amplifications. PCR products were purified using the GRS PCR & Gel Band Purification Kit (GRiSP, Porto, Portugal) according to the manufacturer’s instructions. Sequencing was carried out using the same primers employed for amplification. Raw sequences were verified and edited using MEGA 12 software and aligned using the CLUSTALW algorithm. Species identification was based on BLAST comparisons against GenBank, MycoBank, Fusarium ID, and CBS-KNAW databases. Only sequences with ≥99% similarity were accepted for definitive species-level identification. Phylogenetic analyses were performed using the Maximum Likelihood method in MEGA12 (Appendix A). Evolutionary distances were calculated using the Kimura (1980) 2-parameter model [54], and the robustness of the tree topology was evaluated through bootstrap analysis with 500 or 1000 replicates [55].

## Figures and Tables

**Figure 1 toxins-17-00468-f001:**
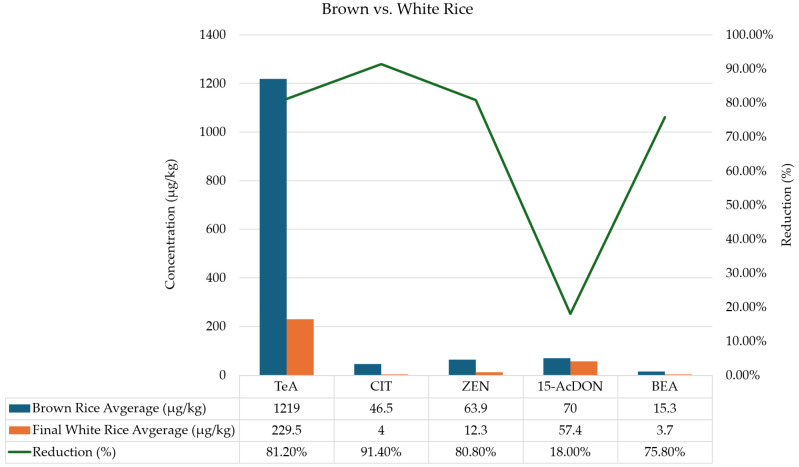
Comparison of mycotoxin levels (average of the 2 varieties) in brown vs. white rice. The bars represent the concentration (left axis) of the brown rice and the final white rice. In contrast, the line represents the percentage of reduction (right axis) between the brown rice and the final white rice, the two commonly consumed rice types.

**Figure 2 toxins-17-00468-f002:**
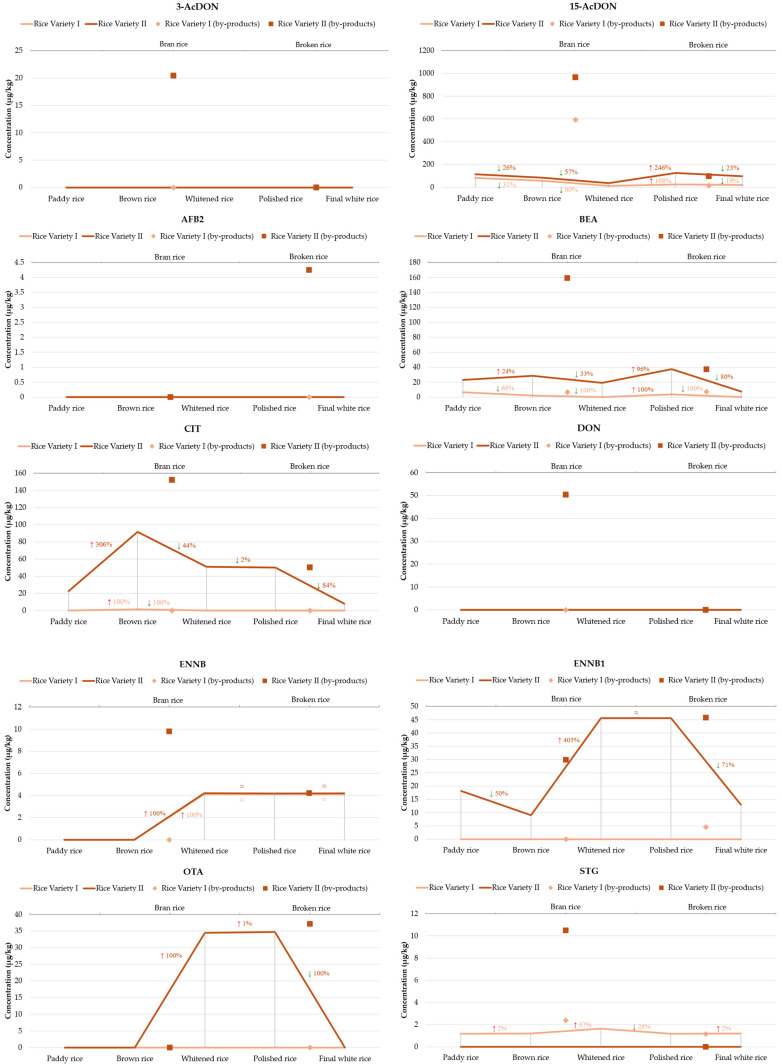
Evolution of positive mycotoxins through the rice processing steps. 3-AcDON: 3-acetyl-deoxynivalenol; 15-AcDON: 15-acetyl-deoxynivalenol; AFB2: aflatoxin B2; BEA: beauvericin; CIT: citrinin; DON: deoxynivalenol; ENNB: enniatin B; ENNB1: enniatin B1; OTA: ochratoxin A; STG: sterigmatocystin; TeA: tenuazonic acid; T-2: T-2 toxin; ZEN: zearalenone.

**Figure 3 toxins-17-00468-f003:**
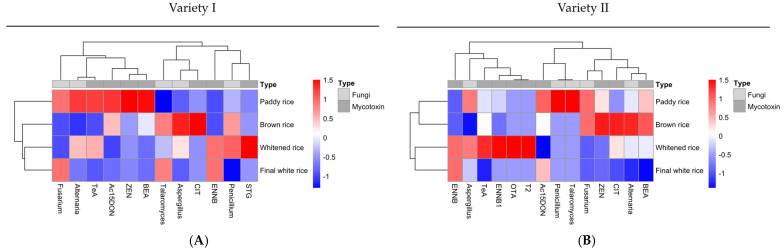
Clustered heatmap of fungi isolated and mycotoxin concentrations across rice processing stages for variety I (**A**) and variety II (**B**). The heatmap displays standardized values (z-scores) for fungal isolates and detected mycotoxins across four sample types. Rows correspond to variables (fungi and mycotoxins), and columns represent sample types. Hierarchical clustering was performed on both rows and columns to group similar contamination profiles, with accompanying dendrograms indicating relative similarity. The color scale ranges from blue (values below the mean) through white (mean levels) to red (values above the mean), reflecting standardized deviations from the mean for each variable. Clusters of high values (red) highlight sample types with elevated fungal loads or mycotoxin concentrations. Annotations above the heatmap differentiate fungal variables (light gray bar) from mycotoxin variables (dark gray bar).

**Table 1 toxins-17-00468-t001:** Mycotoxins amounts in screened samples. Values include concentration (µg/kg) ± SD (*n* = 4), average (μg/kg), minimum (max, μg/kg) and maximum (min, μg/kg).

**Rice Variety I**										
Mycotoxins	Paddy rice	Brown rice	Rice bran	Whitened rice	Polished rice	Broken rice	Final white rice	Average (µg/kg)	Min (µg/kg)	Max (µg/kg)
3-AcDON	nd	nd	nd	nd	nd	nd	nd	-	-	-
15-AcDON	81.15 ± 5.60	56.21 ± 0.71	591.58 ± 4.20	11.34 ± 2.20	23.54 ± 4.24	13.72 ± 2.07	19.04 ± 0.33	113.80	11.34	591.58
AFB1	nd	nd	nd	nd	nd	nd	nd	-	-	-
AFB2	nd	nd	nd	nd	nd	nd	nd	-	-	-
AFG1	nd	nd	nd	nd	nd	nd	nd	-	-	-
AFG2	nd	nd	nd	nd	nd	nd	nd	-	-	-
BEA	6.58 ± 0.30	2.10 ± 0.02	6.62 ± 0.76	nd	3.70 ± 0.01	7.37 ± 0.32	nd	5.27	2.10	7.37
CIT	nd	1.39 ± 0.20	nd	nd	nd	nd	nd	-	1.39	1.39
CPZ	nd	nd	nd	nd	nd	nd	nd	-	-	-
DON	nd	nd	nd	nd	nd	nd	nd	-	-	-
ENNA	nd	nd	nd	nd	nd	nd	nd	-	-	-
ENNA1	nd	nd	nd	nd	nd	nd	nd	-	-	-
ENNB	≤LOQ	≤LOQ	≤LOQ	4.13 ± 0.01	4.14 ± 0.01	4.15 ± 0.03	4.13 ± 0.02	4.14	4.13	4.15
ENNB1	nd	≤LOQ	≤LOQ	nd	nd	4.57 ± 0.01	nd	-	4.57	4.57
FB1	nd	nd	nd	nd	nd	nd	nd	-	-	-
FB2	nd	nd	nd	nd	nd	nd	nd	-	-	-
HT-2	nd	nd	nd	nd	nd	nd	nd	-	-	-
OTA	nd	nd	nd	nd	nd	nd	nd	-	-	-
STG	1.18 ± 0.03	1.20 ± 0.02	2.39 ± 0.14	1.64 ± 0.01	1.18 ± 0.02	1.17 ± 0.01	1.20 ± 0.01	1.42	1.17	2.39
TeA	292.10 ± 8.09	42.94 ± 9.76	917.00 ± 14.65	217.99 ± 19.8	79.55 ± 9.91	323.59 ± 4.07	51.30 ± 18.73	274.92	42.94	917.00
T-2	nd	nd	3.32 ± 0.52	nd	nd	nd	nd	-	3.32	3.32
ZEN	55.14 ± 9.39	11.57 ± 1.02	200.91 ± 13.09	9.92 ± 1.85	10.90 ± 1.45	9.76 ± 0.38	9.10 ± 0.82	43.90	9.1	200.91
Total *	5	6	6	5	6	7	5			
**Rice Variety II**										
Mycotoxins	Paddy rice	Brown rice	Rice bran	Whitened rice	Polished rice	Broken rice	Final white rice	Average (µg/kg)	Min (µg/kg)	Máx (µg/kg)
3-AcDON	nd	nd	20.44 ± 2.94	nd	nd	nd	nd	-	20.44	20.44
15-AcDON	113.28 ± 16.23	83.75 ± 10.20	965.23 ± 14.72	35.86 ± 3.91	124.25 ± 4.08	95.29 ± 11.88	95.74 ± 10.09	216.20	35.86	965.23
AFB1	nd	nd	nd	nd	nd	nd	nd	-	-	-
AFB2	nd	nd	nd	nd	nd	4.25 ± 0.82	nd	-	4.25	4.25
AFG1	nd	nd	nd	nd	nd	nd	nd	-	-	-
AFG2	nd	nd	nd	nd	nd	nd	nd	-	-	-
BEA	22.99 ± 0.05	28.52 ± 0.57	159.19 ± 0.84	19.16 ± 1.92	37.57 ± 0.43	37.30 ± 0.23	7.41 ± 0.03	44.59	7.41	159.19
CIT	22.57 ± 1.28	91.61 ± 0.40	152.23 ± 0.99	50.95 ± 0.92	50.10 ± 0.05	50.21 ± 0.04	7.99 ± 0.07	60.81	7.99	152.23
CPZ	nd	nd	nd	nd	nd	nd	nd	-	-	-
DON	nd	nd	50.32 ± 5.57	nd	nd	nd	nd	-	50.32	50.32
ENNA	nd	nd	nd	nd	nd	nd	nd	-	-	-
ENNA1	nd	nd	nd	nd	nd	nd	nd	-	-	-
ENNB	≤LOQ	≤LOQ	9.80 ± 0.08	4.21 ± 0.09	4.19 ± 0.02	4.22 ± 0.04	4.20 ± 0.06	5.32	4.19	9.80
ENNB1	18.21 ± 0.88	9.03 ± 0.02	29.90 ± 0.08	45.61 ± 0.02	45.62 ± 0.01	45.78 ± 0.33	13.01 ± 1.94	29.59	9.03	45.78
FB1	nd	nd	nd	nd	nd	nd	nd	-	-	-
FB2	nd	nd	nd	nd	nd	nd	nd	-	-	-
HT-2	nd	nd	nd	nd	nd	nd	nd	-	-	-
OTA	nd	nd	nd	34.44 ± 0.09	34.68 ± 0.09	37.11 ± 0.06	nd	35.41	34.44	37.11
STG	nd	nd	10.49 ± 0.14	nd	nd	nd	nd	-	10.49	10.49
TeA	2121.37 ± 5.10	2395.11 ± 3.51	5419.88 ± 4.74	4264.81 ± 6.42	2219.83 ± 4.41	2812.20 ± 13.05	407.76 ± 14.92	2805.85	407.76	5419.88
T-2	nd	nd	2.69 ± 0.38	4.61 ± 0.55	nd	nd	nd	3.65	2.69	4.61
ZEN	62.98 ± 0.56	116.31 ± 15.97	220.45 ± 7.91	17.41 ± 2.93	17.35 ± 3.11	16.64 ± 0.39	15.44 ± 0.29	66.62	15.44	220.45
Total *	6	6	11	9	8	9	7			

SD: standard deviation; LOQ: Limit of quantification; nd—not detected; 3-AcDON: 3-acetyl-deoxynivalenol; 15-AcDON: 15-acetyl-deoxynivalenol; AFB1: aflatoxin B1; AFB2: aflatoxin B2; AFG1: aflatoxin G1; AFG2: aflatoxin G2; BEA: beauvericin; CIT: citrinin; CPZ: cyclopiazonic acid; DON: deoxynivalenol; ENNA: enniatin A; ENNA1: enniatin A1; ENNB: enniatin B; ENNB1: enniatin B1; FB1: fumonisin B1; FB2: fumonisin B2; HT-2: HT-2 toxin; OTA: ochratoxin A; STG: sterigmatocystin; TeA: tenuazonic acid; T-2: T-2 toxin; ZEN: zearalenone; * Total: correspond to the total co-occurring mycotoxins per sample.

**Table 2 toxins-17-00468-t002:** Strain identification (ID) of the fungi isolated from the rice varieties using morphological and molecular methods.

**Rice Variety I**						
		Samples				
Morphological ID	Molecular ID	Paddy rice	Brown rice	Whitened rice	Final white rice	Total *n* (%)
*Aspergillus* spp.		1	5	3	1	10 (20.8)
	*A. fumigatus*	1				
	*A. chevalieri*		2			
	*A. porosus*		2			
	*A. pseudoglaucus*		1			
	*A. clavatus*			2		
	*A. montevidensis*			1		
	*A. flavus*				1	
*Penicillium* spp.		4	7	8	1	20 (41.7)
	*P. charlesii*	1	2	7		
	*P. viridicatum*	2		1		
	*P. citrinum*	1				
	*P. polonicum*		1			
	*P. sajarovii*		2			
	*P. echinulonalgiovense*		2			
	*P. brevicompactum*				1	
*Talaromyces* spp.		0	0	1	2	3 (6.3)
	*T. wortmannii*			1		
	*T. rugulosus*				2	
*Fusarium* sp.		1	0	0	1	2 (4.2)
	*F. fujikuroi*	1			1	
*Alternaria* spp.		6	2	4	1	13 (27.1)
	*A. resedae*	1	1	1		
	*A. alternata*	3	1	1		
	*A. destruens*	1				
	*A. jacinthicola*	1				
	*A. tenuissima*			1		
	*A. arborescens*			1	1	
Total *n* (%)		12 (25.0)	14 (29.2)	16 (33.3)	6 (12.5)	48
**Rice Variety II**						
		Samples				
Morphological ID	Molecular ID	Paddy rice	Brown rice	Whitened rice	Final white rice	Total *n* (%)
*Aspergillus* spp.		2	0	2	1	5 (18.5)
	*A. fumigatus*	2				
	*A. chevalieri*			1		
	*A. neotritici*			1	1	
*Penicillium* sp.		2	0	0	0	2 (7.4)
	*P. viridicatum*	2				
*Talaromyces* sp.		1	0	0	0	1 (3.7)
	*T. rugulosus*	1				
*Fusarium* sp.		1	1	0	0	2 (7.4)
	*F. tanahbumbuense*	1	1			
*Alternaria* spp.		4	9	4	0	17 (63.0)
	*A. resedae*	1				
	*A. alternata*	3	4	3		
	*A. destruens*		4			
	*A. longipes*		1			
	*A. arborescens*			1		
Total *n* (%)		10 (37.0)	10 (37.0)	6 (22.2)	1 (3.7)	27

## Data Availability

The data presented in this study are available on request from the corresponding author. (Authors are going to explore further function and the data will be available on request).

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
