# Peer review of "Integrated Assessment of Fungi Contamination and Mycotoxins Levels Across the Rice Processing Chain"

_toxins, 2025, doi:10.3390/toxins17090468_

Round 1

Reviewer 1 Report

Comments and Suggestions for Authors

In this manuscript,the fungi and mycotoxins in rice are detected. It is useful for the food safety. There are some points need to be revised.

This study investigated the occurrence of fungi and mycotoxins throughout the rice processing chain, from paddy rice to final white rice, in two rice varieties. A total of 75 fungal isolates were identified, belonging to the genera Penicillium, Alternaria, Aspergillus, Fusarium, and Talaromyces. Variety I exhibited a higher prevalence of Penicillium and Alternaria, whereas Variety II was dominated mainly by Alternaria, accounting for 63% of all isolates. Multi-mycotoxin screening revealed contamination by tenuazonic acid (TeA), zearalenone (ZEN), and 15-acetyl-deoxynivalenol (15-AcDON), with TeA concentrations exceeding 4000µg/kg in whitened rice of variety II. Contamination profiles show paddy and brown rice samples clustering together due to elevated fungal loads and higher mycotoxin levels. In contrast, whitened and final white rice clustered separately, indicating an effective reduction in fungal load, yet the persistence of specific toxins, such as TeA, 15-AcDON, ZEN, and CIT. The co-clustering of Alternaria with TeA and ZEN suggests strong field-related contamination sources. Despite significant reductions in microbial contamination through processing, residual mycotoxins highlight an incomplete mitigation of food safety risks.

These findings underscore the need for comprehensive surveillance and integrated management practices across the rice supply chain to minimize potential health hazards associated with fungal contaminants and their toxic metabolites. Studying contamination in the rice processing chain is important for food safety. The research perspective is significantly innovative, and the research results have reference value for improving the level of rice processing. Some of the results fill the gap in rice processing in terms of toxin accumulation. The research design is reasonable, and the diagrams are appropriate. I suggest to accept it.

  1. In figure 1, I suggest change the line to column. For the abscissa is discontinuous.
  2. The references need to checked carefully. Some references give all the authors, but some just one author and et. al.
  3. What’s variety I and variety II? Its Japonica rice or indica rice?

Author Response

General comment:

In this manuscript the fungi and mycotoxins in rice are detected. It is useful for the food safety. There are some points need to be revised.

This study investigated the occurrence of fungi and mycotoxins throughout the rice processing chain, from paddy rice to final white rice, in two rice varieties. A total of 75 fungal isolates were identified, belonging to the genera Penicillium, Alternaria, Aspergillus, Fusarium, and Talaromyces. Variety I exhibited a higher prevalence of Penicillium and Alternaria, whereas Variety II was dominated mainly by Alternaria, accounting for 63% of all isolates. Multi-mycotoxin screening revealed contamination by tenuazonic acid (TeA), zearalenone (ZEN), and 15-acetyl-deoxynivalenol (15-AcDON), with TeA concentrations exceeding 4000µg/kg in whitened rice of variety II. Contamination profiles show paddy and brown rice samples clustering together due to elevated fungal loads and higher mycotoxin levels. In contrast, whitened and final white rice clustered separately, indicating an effective reduction in fungal load, yet the persistence of specific toxins, such as TeA, 15-AcDON, ZEN, and CIT. The co-clustering of Alternaria with TeA and ZEN suggests strong field-related contamination sources. Despite significant reductions in microbial contamination through processing, residual mycotoxins highlight an incomplete mitigation of food safety risks.

These findings underscore the need for comprehensive surveillance and integrated management practices across the rice supply chain to minimize potential health hazards associated with fungal contaminants and their toxic metabolites. Studying contamination in the rice processing chain is important for food safety. The research perspective is significantly innovative, and the research results have reference value for improving the level of rice processing. Some of the results fill the gap in rice processing in terms of toxin accumulation. The research design is reasonable, and the diagrams are appropriate. I suggest to accept it.

Response to general comment: We thank the reviewer for their pleasant comments and valuable suggestions, which significantly improved our study.

Comments 1: In Figure 1, I suggest change the line to column. For the abscissa is discontinuous.

Response 2: We appreciate the reviewer’s suggestion. In Figure 1, the bars represent the concentration (left axis) of the brown rice and the final white rice. In contrast, the line represents the percentage of reduction (right axis) between the brown rice and the final white rice, the two commonly consumed rice types. To make a better distinction between concentration and percentage of reduction, we propose bar and line charts, respectively. We have improved the figure caption to clarify this issue (lines 224-227). However, if you deem it necessary, we are available to make modifications.

Comments 2: The references need to checked carefully. Some references give all the authors, but some just one author and et. al..

Response 2: We have accordingly revised and updated the manuscript in the reference section (Lines 738-740).

Comments 3: What’s variety I and variety II? Its Japonica rice or indica rice?

Response 3: We agree with the reviewer; however, we received blind samples from the company, and they did not provide the necessary information.

Reviewer 2 Report

Comments and Suggestions for Authors

Title: The current title is not adequate and should be revised to better reflect the actual work that has been conducted.

Abstract: The terms Variety I and Variety II are not explained, yet they are used in the abstract, which may confuse readers. Please provide a brief explanation. The number of mycotoxins screened should be explicitly stated. The sentence in lines 16–18 is unclear and should be rephrased. Overall, the abstract could be improved by providing a clearer and more concise summary of the main findings.

Introduction: In line 52, only source number 9 is cited. This is insufficient, as there is already a substantial body of literature on mycotoxins in rice. Please include additional references. Older but relevant studies have not been addressed and should also be incorporated to provide a more comprehensive background.

Methods and Results: The part "Method validation" currently appears in the Results and Discussion section. This should be moved to the Methods section. The use of the word demonstrated may be too strong, as other processes could potentially have contributed to the observed decrease in detectable mycotoxins. Consider softening the statement to reflect this uncertainty.

Figures: In Figure 1, it would be helpful to include the official EU regulatory limits as a reference line. This would allow readers to easily assess whether any mycotoxin levels exceed the permissible thresholds.

Table: Please add spp. instead of sp. when more than one species of a genus was isolated. If only one species was isolated, sp. remains appropriate.

Author Response

Comments 1: Title: The current title is not adequate and should be revised to better reflect the actual work that has been conducted.

Response 1: We agree with the reviewer. The title was changed to: Integrated Assessment of Fungi Contamination and Mycotoxins Levels Across the Rice Processing Chain.

Comments 2: Abstract: The terms Variety I and Variety II are not explained, yet they are used in the abstract, which may confuse readers. Please provide a brief explanation. The number of mycotoxins screened should be explicitly stated. The sentence in lines 16–18 is unclear and should be rephrased. Overall, the abstract could be improved by providing a clearer and more concise summary of the main findings.

Response 2: We have accordingly changed to emphasize this point and improve the abstract for better understanding. On line 7, it was stated the variety I and II. In line 11, the number of screened mycotoxins was added (22). Sentences 14-19 were revised to enhance clarity and provide a concise summary of the main findings.

Comments 3: Introduction: In line 52, only source number 9 is cited. This is insufficient, as there is already a substantial body of literature on mycotoxins in rice. Please include additional references. Older but relevant studies have not been addressed and should also be incorporated to provide a more comprehensive background.

Response 3: Thank you for bringing this to our attention. We have followed the suggestion and added more references (4,9-13) to the Introduction section (line 52).

Comments 4: Methods and Results: The part “Method validation” currently appears in the Results and Discussion section. This should be moved to the Methods section. The use of the word demonstrated may be too strong, as other processes could potentially have contributed to the observed decrease in detectable mycotoxins. Consider softening the statement to reflect this uncertainty.

Response 4: Thank you for bringing this to our attention. Mycotoxins were extracted using a modified QuEChERS-based method, adapted from previously established protocols [41, 42]. We had to validate this protocol in our matrix, and we include the results from that validation in the Results and Discussion section. However, if you deem it necessary, we are available to make modifications.

Comments 5: Figures: In Figure 1, it would be helpful to include the official EU regulatory limits as a reference line. This would allow readers to easily assess whether any mycotoxin levels exceed the permissible thresholds.

Response 5: Thank you for bringing this to our attention. For the mycotoxins displayed in Figure 1, there are no official EU regulatory limits for rice. The EU has not established maximum limits for ZEN in rice, only set for other cereals, 100 µg/kg (for unprocessed grains) and 75 µg/kg (for grains intended for consumption). However, that comparison is addressed in this manuscript at Lines 318-321.

Comments 6: Table: Please add spp. instead of sp. when more than one species of a genus was isolated. If only one species was isolated, sp. remains appropriate.

Response 6: We appreciate the reviewer’s correction. The manuscript was corrected to “spp” (Section 2.3. Mycoflora Isolation and Identification, Table 3).

Reviewer 3 Report

Comments and Suggestions for Authors

Dear editor,

The work is well structured and the bibliographic references are consistent with the content of the text. Therefore, this reviewer suggests that this study is potentially suitable for publication in Toxins. The manuscript is acceptable for publication if some minor revisions are made to improve the quality of the content and writing. Some comments on the content are:

Line 16. Compared to the other mycotoxins mentioned in the abstract, CIT is indicated here for the first time; please correct by inserting its full name.

Lines 31-37. This explanation is redundant and overly didactic; therefore, I would suggest removing it.

Lines 71-72. UHPLC-MS/MS: First, write the full name, followed by the acronym in parentheses.

Line 95. It is recommended to move Table 1 to the supporting information file.

Lines 124-128. Please include bibliographic references to support this statement.

Line 544. Please specify the column geometry as follows: (100 × 2.1 mm L * ID).

Line 544.  What is the reason for setting the column temperature at 28°C?

Line 554. Why was nitrogen used as the collision gas instead of argon?

Lines 584-586. Insert bibliographic references, such as:

  • Visagie, C.M., Houbraken, J., Frisvad, J.C., Hong, S.B., Klaassen, C.H., Perrone, G., Seifert, K.A., Varga, J., Yaguchi, T., Samson, R.A., 2014. Identification and nomenclature of the genus Penicillium. Stud. Mycol. 78, 343–371. https://doi.org/10.1016/j.simyco.2014.09.001
  • Marco Garello, Edoardo Piombo, Fabio Buonsenso, Simona Prencipe, Silvia Valente, Giovanna Roberta Meloni, Marina Marcet-Houben, Toni Gabaldón, Davide Spadaro, Several secondary metabolite gene clusters in the genomes of ten Penicillium spp. raise the risk of multiple mycotoxin occurrence in chestnuts, Food Microbiology, Volume 122, 2024, 104532, https://doi.org/10.1016/j.fm.2024.104532
  • Acharya, Tankeshwar, and Janelle Hare. "Sabouraud agar and other fungal growth media." Laboratory protocols in fungal biology: current methods in fungal biology. Cham: Springer International Publishing, 2022. 69-86. https://doi.org/10.1007/978-3-030-83749-5_2

Line 453. Have you considered or evaluated the effectiveness of specific treatment methods or processing technologies that could further reduce the residual presence of mycotoxins in the final products? If so, what are the results, and if not, what strategies would you suggest to mitigate this residual risk?

Please check the formatting of the supporting information, as Figure 2 appears cropped and is not fully visible on the page.

Author Response

General comment: The work is well structured and the bibliographic references are consistent with the content of the text. Therefore, this reviewer suggests that this study is potentially suitable for publication in Toxins. The manuscript is acceptable for publication if some minor revisions are made to improve the quality of the content and writing.

Response to general comment: We thank the reviewer for their pleasant comments and valuable suggestions, which significantly improved our study.

Comments 1: Line 16. Compared to the other mycotoxins mentioned in the abstract, CIT is indicated here for the first time; please correct by inserting its full name.

Response 1: We thank the reviewer for the correction. The manuscript was corrected to “citrinin (CIT)” (line 16).

Comments 2: Lines 31-37. This explanation is redundant and overly didactic; therefore, I would suggest removing it.

Response 2: Thank you for the comment. We review the paragraph and change it to enhance clarity and provide a concise explanation of the origin and processing steps of the samples that we studied.

Comments 3: Lines 71-72. UHPLC-MS/MS: First, write the full name, followed by the acronym in parentheses.

Response 3: We thank the reviewer for the correction. The manuscript was corrected to “Ultra-high performance liquid chromatography tandem mass spectrometry (UHPLC-MS/MS)” (line 72-73).

Comments 4: Line 95. It is recommended to move Table 1 to the supporting information file.

Response 4: We agree with the reviewer and have made the necessary changes. We also revised the numbering of the order table after removing table 1 (marked in red in the manuscript).

Comments 5: Lines 124-128. Please include bibliographic references to support this statement.

Response 5: We appreciate the reviewer’s correction. In the manuscript, references 6-8 were added to support the statement (line 129).

Comments 6: Line 544. Please specify the column geometry as follows: (100 × 2.1 mm L * ID).

Response 6: Thank you for bringing this to our attention. We have changed in the manuscript (Line 539).

Comments 7: Line 544. What is the reason for setting the column temperature at 28°C?

Response 7: Thank you for the comment. We establish this temperature to minimize environmental variation and optimize separation without compromising column or analyte stability.

Comments 8: Line 554. Why was nitrogen used as the collision gas instead of argon?

Response 8: Thank you for bringing this to our attention. Agilent triple quadrupole (QQQ) mass spectrometers can use nitrogen in both the collision cell and as a drying/sheath gas in electrospray ionization. To ensure consistent performance, Agilent QQQ systems typically require two independent nitrogen supplies, one dedicated to the collision cell and one to the nebulizer/drying functions. High-purity nitrogen (>99.9995%) at stable pressures and flow rates is essential, and we have dedicated nitrogen generators to meet these requirements. Nitrogen enables efficient but controlled ion fragmentation and improved desolvation through Agilent’s Jet Stream technology. It is also cheaper, easier to supply, and more compatible with high-vacuum conditions than argon, which can increase background pressure and reduce sensitivity.

Comments 9: Line 584-586. Insert bibliographic references, such as:

Visagie, C.M., Houbraken, J., Frisvad, J.C., Hong, S.B., Klaassen, C.H., Perrone, G., Seifert, K.A., Varga, J., Yaguchi, T., Samson, R.A., 2014. Identification and nomenclature of the genus Penicillium. Stud. Mycol. 78, 343–371. https://doi.org/10.1016/j.simyco.2014.09.001

Marco Garello, Edoardo Piombo, Fabio Buonsenso, Simona Prencipe, Silvia Valente, Giovanna Roberta Meloni, Marina Marcet-Houben, Toni Gabaldón, Davide Spadaro, Several secondary metabolite gene clusters in the genomes of ten Penicillium spp. raise the risk of multiple mycotoxin occurrence in chestnuts, Food Microbiology, Volume 122, 2024, 104532, https://doi.org/10.1016/j.fm.2024.104532

Acharya, Tankeshwar, and Janelle Hare. “Sabouraud agar and other fungal growth media.” Laboratory protocols in fungal biology: current methods in fungal biology. Cham: Springer International Publishing, 2022. 69-86. https://doi.org/10.1007/978-3-030-83749-5_2

Response 9: Thank you for the comment. We added some of the indicated bibliographic references to the reference list (Line 583).

Comments 10: Line 453. Have you considered or evaluated the effectiveness of specific treatment methods or processing technologies that could further reduce the residual presence of mycotoxins in the final products? If so, what are the results, and if not, what strategies would you suggest to mitigate this residual risk?

Response 10: Thank you for bringing this to our attention. In this study, we did not evaluate extra decontamination steps beyond the industrial chain (de-husking, whitening, polishing). Milling and polishing clearly lowered fungal loads and most mycotoxins in the edible fraction, but several toxins persisted and concentrated in by-products (bran, broken rice).

Future strategies to mitigate residual mycotoxin risk in rice should focus on innovative post-harvest decontamination technologies, combined with optimized milling practices. Promising approaches include ozone treatment, which has shown effectiveness in degrading zearalenone and other toxins with limited impact on grain quality; cold atmospheric plasma, capable of inactivating fungi and reducing aflatoxins, OTA, and trichothecenes in cereals; and electron-beam irradiation, which can significantly lower ZEN and OTA levels while preserving nutritional value when doses are carefully controlled. Together with preventive measures, such as moisture control, hermetic storage, and kernel sorting, these technologies could offer a multi-layered approach to reduce consumer exposure. However, future research should also verify the toxicity of degradation products and adapt these methods for industrial-scale application. Indeed, it is an important approach that could be done and will be considered in future work.

Bibliographic support:

Chandravarnan, P., D. Agyei, and A. Ali, Green and sustainable technologies for the decontamination of fungi and mycotoxins in rice: A review. Trends in Food Science & Technology, 2022. 124: p. 278-295.

Furlong, E.B., et al., Mitigation of Mycotoxins in Food-Is It Possible? Foods, 2024. 13(7).

Pandey, A.K., et al., Fungal mycotoxins in food commodities: present status and future concerns. Frontiers in Sustainable Food Systems, 2023. Volume 7 - 2023.

Naeem, I., et al., Aflatoxins in the rice production chain: A review on prevalence, detection, and decontamination strategies. Food Research International, 2024. 188: p. 114441.

Karlovsky, P., et al., Impact of food processing and detoxification treatments on mycotoxin contamination. Mycotoxin Research, 2016. 32(4): p. 179-205.

Comments 11: Please check the formatting of the supporting information, as Figure 2 appears cropped and is not fully visible on the page.

Response 11: Thank you for bringing this to our attention. The formatting of the supporting information was checked and corrected.
